# Biodiversity of β-Carboline Profile of *Banisteriopsis caapi* and Ayahuasca, a Plant and a Brew with Neuropharmacological Potential

**DOI:** 10.3390/plants9070870

**Published:** 2020-07-09

**Authors:** Beatriz Werneck Lopes Santos, Regina Célia de Oliveira, Julia Sonsin-Oliveira, Christopher William Fagg, José Beethoven Figueiredo Barbosa, Eloisa Dutra Caldas

**Affiliations:** 1Laboratory of Toxicology, Department of Pharmacy, Faculty of Health Sciences, University of Brasilia, Brasilia-DF 70910-900, Brazil; beatrizwerneck.bia@hotmail.com; 2Department of Botany, Institute of Biology, University of Brasilia, Brasilia-DF 70910-900, Brazil; reginacelia@unb.br (R.C.d.O.); jsonsin@yahoo.com.br (J.S.-O.); 3Faculty of Ceilândia, University of Brasilia, Brasilia-DF 72220-275, Brazil; acaciafagg@gmail.com; 4Department of Phytotechnology, Federal University of Roraima, Boa Vista-Roraima 69310-000, Brazil; vdiretorplantio_pesquisa.dg@udv.org.br; 5Centro Espírita Beneficente União do Vegetal (UDV), Brasília-DF 70040-904, Brazil

**Keywords:** ß-carbolines, DMT, *Banisteriopsis caapi*, ayahuasca, Brazil

## Abstract

Ayahuasca is a psychoactive infusion with a large pharmacological application normally prepared with *Banisteriopsis caapi*, which contains the monoamine oxidase inhibitors β-carbolines, and *Psichotria virids*, which contains the serotonin receptor agonist N,N dimethyltryptamine (DMT). The objectives of this study were to investigate the chemical profile of *B. caapi* and of ayahuasca collected in various Brazilian regions. In total, 176 plant lianas, of which 159 *B.*
*caapi* and 33 ayahuasca samples were analyzed. Dried liana samples were powdered, extracted with methanol, diluted, and analyzed by LC-MS/MS. Ayahuasca samples were diluted and analyzed. Mean concentrations in *B. caapi* were 4.79 mg/g harmine, 0.451 mg/g harmaline, and 2.18 mg/g tetrahydroharmine (THH), with a high variability among the samples (RSD from 78.9 to 170%). Native *B. caapi* samples showed significantly higher harmine concentrations than cultivated ones, and samples from the Federal District/Goiás had higher THH content than those collected in the State of Acre. The other Malpighiaceae samples did not contain β-carbolines, except for one *D. pubipetala* sample. Concentrations in ayahuasca samples ranged from 0.109 to 7.11 mg/mL harmine, 0.012 to 0.945 mg/mL harmaline, 0.09 to 3.05 mg/mL THH, and 0.10 to 3.12 mg/mL DMT. The analysis of paired ayahuasca/*B. caapi* confirmed that harmine is reduced to harmaline and to THH during the brew preparation. This is the largest study conducted with Malpighiaceae samples and showed a large variability in the main β-carbolines present in *B. caapi*. This biodiversity is a challenge for standardization of the material used in ethnopharmacological studies of *B.*
*caapi* and ayahuasca.

## 1. Introduction

Ayahuasca is a psychotropic beverage traditionally used in religious rituals and as medicine among indigenous communities from the Amazonian region, a use that has expanded since the 1970s to all Brazilian regions and more recently to other countries [1,2]. The beverage is usually prepared by the decoction of the liana *Banisteriopsis caapi* (Spruce ex Griseb.) C.V. Morton and *Psychotria viridis* Ruiz & Pav [3,4] and may be called by other names, such as vegetal and Daime [1].

*Banisteriopsis caapi* contains β-carboline alkaloids, mainly harmine, harmaline, and tetrahydroharmine (THH) (Figure 1), which are inhibitors of monoamine oxidases (MAO) [5,6]. *Psychotria viridis* has the hallucinogenic indole N,N-dimethyltryptamine (DMT; Figure 1), a serotonin agonist that causes intense cognitive and perceptive modifications when administered parenterally in humans [7]. When taken orally, DMT is oxidized by MAO, a process that is prevented by the presence of β-carbolines after ayahuasca ingestion, allowing the DMT to reach the serotoninergic receptors in the nervous system causing the psychotropic effects of the beverage [4,5,8,9]. Harmine and harmaline are also shown to be hallucinogenic in humans, probably due to their interaction with serotonin, dopamine, benzodiazepine, imidazoline, and opiate receptors, acting as either agonists or antagonists [10,11,12]. The activity of THH in ayahuasca is apparently more a function of its inhibition of serotonin uptake than of its action as a MAO inhibitor [13].

The therapeutic potential of ayahuasca, *B. caapi*, and its components, mainly harmine, was widely explored in the last 20 years [14]. The β-carbolines present in *B. caapi* were shown to stimulate neurogenesis in vitro and the modulation of brain plasticity could contribute to the antidepressant effects of ayahuasca [15] and harmine [16,17]. *B. caapi* extracts showed to act against neurodegenerative diseases, such as Parkinson’s disease [18,19,20,21]. Furthermore, various studies have investigated the therapeutic potential of ayahuasca for depression and addiction in experimental animals [22,23,24,25] and humans [26,27,28].

The species *B. caapi* is an Amazon native giant liana from the Malpighiaceae family [4,29,30], and ayahuasca brewers recognize different “types” of liana. Langdon [31] has documented at least 18 types of *B. caapi* by the Siona Indigenous people, highlighting features such as length and width of the stem, size and shape of the leaves, and the presence or absence of flowers. Schultes et al. [32] remarked that Amazonian natives acknowledged at least 30 different types of *B. caapi,* which all have different uses in their traditional medicine. Peruvian brewers (ayahuasqueros) claim to distinguish as many as 10 types of *B. caapi* lianas [4].

Some studies have investigated the phytochemistry of the species used in the preparation of the ayahuasca beverage; however, only few samples were analyzed [4,21,32,33,34], which limits the possibility to capture the diversity of the chemical profile of the plants. The aim of this study was to make a thorough chemical investigation of the β-carbolines harmine, harmaline, and tetrahydroharmine levels present in *B. caapi* and in ayahuasca brew in samples collected from various regions in Brazil. Additionally, other plants used in the preparation of the brew were also analyzed, and the levels of DMT were investigated in the brew.

## 2. Material and Methods

### 2.1. Banisteriopsis spp., Diplopterys pubipetala, and Ayahuasca Brew Samples

A total of 176 liana samples were collected, 159 of *B. caapi* and 17 samples of different species that are also used in ayahuasca preparations by some groups (Schultes et al., 2001): four samples of *B. laevifolia* (A.Juss.) B. Gates, one of *B. megaphylla* (A.Juss.) B.Gates, one of *B. muricata* (Cav.) Cuatrec., one of *B. oxyclada* (A.Juss.) B. Gates, one of *B. variabilis* B.Gates, one of *B. gardneriana* (A.Juss.) W.R.Anderson & B.Gates, three of uncharacterized *Banisteriopsis* spp., and five of *Diplopterys pubipetala* (A.Juss.) W.R.Anderson & C.C.Davis.

Liana samples were collected from 2016 to 2019 in different states in the northern region of Brazil (Acre (AC), Amazonas (AM), Rondônia (RO), and Pará (PA)) and in the mid-western region (Federal District (DF) and Goiás (GO)), where the plants are cultivated (Figure 2). The *B. caapi* samples were classified in four main groups according to botanical characteristics, which were described by the collectors and ayahuasca users: tucunacá, amarelinho, and ourinho (without swollen nodes) and caupuri (characterized by the presence of swollen nodes) (Figure 3). Additionally, seven samples with other descriptions that could be referred to as hybrids were also provided: two samples of caupuri without nodes, one of pajezinho, two tucunacá with nodes, one caboquinho, and one quebrador. Plant samples are stored in the UB (University of Brasilia) herbarium and the UBw wood collection of the University of Brasilia, and detailed information of each sample is shown in Appendix A.

A total of 33 ayahuasca brew samples were provided for analysis: five samples by the Centro Espírita Beneficente União do Vegetal group (UDV), one sample by the Igreja do Culto Eclético da Fluente Luz Universal (ICEFLU), and 27 samples by different groups that were generally referred to in this study as Daime. The beverages were prepared from 2017 to 2020 in the states of Acre and Goiás and the Federal District. Samples were frozen at −20 °C upon arriving at the laboratory until analyzed. Appendix A shows detailed information of each sample.

This work was registered at the National System for the Management of Genetic Heritage and Associated Traditional Knowledge (SisGen) from the Brazilian Ministry of Environment (Protocols A2753ED and A320330).

### 2.2. Chemicals and Reagents

HPLC-grade acetonitrile (ACN) and methanol (MeOH) were purchased from Merck (Darmstadt, Germany); formic acid was obtained from Sigma-Aldrich (St. Louis, MO, USA); ultrapure water obtained through a Milli-Q purification system and the syringe filters used were MillexTM, both from Millipore (Millipore, Bedford, MA, USA). Standards of harmine (98% purity) and harmaline (95% purity) were acquired from Sigma-Aldrich and tetrahydroharmine from LGC (Luckenwalde, Germany; 95% purity). Stock solutions of each standard were prepared at a concentration of 1 mg/mL, from which working solutions at 100, 10, and 1 µg/mL were prepared.

### 2.3. Synthesis of DMT and THH, Identification and Purity Assessment

DMT was synthesized according to Qu et al. [35] with adaptations. In summary, tryptamine (6.25 mmol) was added to a mixture of methanol, acetic acid, and sodium cyanoborohydride (100 mL:1.5 mL:12 mmol), and at 0 °C under a nitrogen atmosphere, a methanol solution of formaldehyde (16.4 mmol) was added dropwise for 90 min. The resulting solution was stirred for 18 h and then alkalinized with sodium carbonate at 2 mol/L until a pH between 8 and 9 was reached. The product was extracted with ethyl acetate and purified through a silica chromatographic column, which yielded a yellowish product. 

When this work was initiated, there was no commercial THH available in the market, and the compound was synthesized following a previously described procedure [36] with some modifications. In summary, a methanolic solution of sodium borohydride (91.2 mg, 2.4 mmol) at 0 °C was added with harmaline (429 mg, 2 mmol). After two hours of reaction, the solution was acidified with 1 mol/L hydrochloric acid and then alkalinized with 5% sodium hydroxide until pH 9.0. The product was extracted with ethyl acetate and then recrystallized with acetone, which yielded white needle-like crystals. The structures of the DMT and THH were confirmed and their purity determined via 1H and 13C NMR experiments in a 600 MHz NMR spectrometer (Bruker^®^, Billerica, MA, USA). The exact masses of the substances were determined by high resolution mass spectrometry with direct injection of a 1 µg/mL methanol solution, with 0.1% formic acid added, in a QTOF spectrometer (Sciex, Framingham, MA, USA). Spectrometric data are reported in the Appendix A.

The purity of the synthetized THH was further confirmed against an analytical standard curve prepared with the analytical LGC standard (95% purity). Seven replicates of a 200 ng/mL solution of the synthetized compound were analyzed, and the results showed that THH was at least 99% pure.

### 2.4. Sample Preparation

Liana samples were dried at room temperature for seven days, powdered by a Wiley mill (Macro Star FT-60, Fortinox, Piracicaba, SP, Brazil), sieved through a 500 µm screen and kept in a plastic bag at room temperature until analysis. Next, 100 mg of each sample were sonicated in 2 mL of methanol and macerated for 24 h at room temperature. The extract was filtered in a 0.45 µm syringe filter and diluted 750-fold with acetonitrile (ACN) and water (1:1). The dilution was adjusted to 100-fold or 10-fold if the sample concentration did not fit within the linear range of the standard analytical curve. Ayahuasca samples were thawed and diluted 750-fold in purified water. In both cases, 1 µL was injected in the HPLC-MS/MS for analyte quantification.

### 2.5. Analysis of the β-carbolines and DMT by Liquid Chromatography–Electrospray Ionization-Tandem Mass Spectrometry (LC-MS/MS)

LC-MS/MS analysis was performed using a Shimadzu LC system (Shimadzu, Kyoto, Japan) coupled to a 4000 Qtrap triple-quadrupole mass spectrometer (Sciex, Framingham, MA, USA), fitted with a Turbo Ion Spray electrospray ionization source (ESI). System operation and data acquisition were controlled by Analyst^®^ (V 1.5.2) software (Sciex).The analyte-dependent MS/MS parameters were optimized by direct infusion of a solution containing all the analytes (10 ng/mL), and the analysis was performed in multiple reaction monitoring (MRM) positive mode. Declustering potential, collision energy, and collision cell exit potential were optimized for the three most abundant transitions for each analyte, and the two most abundant transitions were used as quantifier and qualifier transitions. Ion source parameters were optimized using flow injection analysis of a 5 ng/mL harmaline, and the optimal conditions were: entrance potential at 10 V, ion source at 750 °C, ion source gas 1 and 2 at 50 and 40 psi, ion spray voltage at 4000 V, curtain gas at 12 psi, and collision gas at high. Chromatographic separation was performed with a Luna C18 analytical column (150 × 2 mm, 5 µm) preceded by a C18 security guard cartridge (4.0 × 3.0 mm, 5 µm), both from Phenomenex^®^ (Torrance, CA, USA). The column was kept at 40 °C at a flow rate of 0.4 mL/min, and the injection volume was 1 µL. The mobile phase consisted of 85% water (A) and 15% ACN (B), both added with 0.1% formic acid (12 min run). Appendix A shows the optimized ESI parameters for all analytes.

The method was validated using a *B.* cf. *muricata* liana sample that did not contain β-carbolines, as shown by LC-MS/MS analysis. Selectivity was demonstrated by the absence of any interferences of a *B.* cf. *muricata* extract in the retention time of the analytes. The matrix effect, estimated by the ratio between the average instrument response (areas) of matrix matched standards and neat solution standards (ACN:H_2_O, 1:1), ranged from 2.87 to 6.47% and was considered irrelevant. Furthermore, slopes of the matrix matched, and neat solution standard curves were also not significantly different (*p* > 0.05). The analytical standard curves, prepared in ACN:H_2_O (1:1), showed homoscedastic behavior for all analytes and were linear, with R2 values higher than 0.99. Percent recoveries of the analytes were evaluated by fortifying a sample of *B. muricata* at three different levels (0.75 to 1.5 mg/g) with three to five replicates at each level, and repeatability (same day experiment) and intermediate precision (two different days experiment) were expressed as relative standard deviations of the replicates (%RSD). The results of the validation parameters are shown in Appendix A. The limit of quantification (LOQ) of the method, defined as the lowest level that could be quantified with acceptable recovery (80 to 110%) and precision (RSD < 20%) [37], ranged from 0.0005 mg/mL for harmaline in ayahuasca to 0.75 mg/g for harmine and THH in plant material (Appendix A). The limit of detection (LOD), defined as the lowest concentration that gave a signal-to-noise ratio of three in the LC-MS/MS equipment [38], was 0.0015, 0.0075, and 0.003 mg/g for harmine, harmaline, and THH, respectively, in the plant and 0.00008 and 0.0004 mg/mL for harmine and harmaline, respectively, and 0.0002 mg/mL for THH and DMT in ayahuasca samples.

### 2.6. Statistical Analysis

Concentration data were analyzed using GraphPad Prism 5 (12 March 2007) by one-way analysis of variance (ANOVA) followed by Tukey test. Values are means ± standard error (SEM). In any case, a difference was significant when p was less than 0.05.

## 3. Results

### 3.1. Quantification of β-carbolines in Banisteriopsis spp. and Diplopterys Pubipetala

Table 1 summarizes the β-carboline results for the 159 *B. caapi* samples investigated in this study, and the levels for each sample are shown in Appendix A. The mean levels ranged from 0.451 mg/g for harmaline to 4.79 mg/g for harmine, with THH levels showing the highest variability among the samples (170%). Nine *B. caapi* samples (5.7%) did not contain any of the investigated β-carbolines (<LOD), and no harmine was found in one sample that had the other β-carbolines (Appendix A). In most *B. caapi* samples (88.2%), harmine was present at the highest concentration; eight samples only contained harmine. THH concentrations were higher in 17 samples (11.3%), and harmaline was the analyte with the lowest concentration in most samples (147 samples; 94.7%). The mean level for total β-carbolines levels (harmine + harmaline + THH) was 7.4 mg/g, with an RSD of 40.4% (Table 1). The highest ratios between the β-carbolines in the 141 *B. caapi* samples that contained all compounds (>LOD) were found for THH/harmaline (mean ratio of 5.6), and the highest variability among the ratios was found for THH/harmine (RSD of 331%).

Of the 17 samples from other species used to prepare the ayahuasca beverage and analyzed in this study, only one *D. pubipetala* sample contained β-carbolines (0.16 mg/g harmine, 0.046 mg/g harmaline, and 1.04 mg/g THH; Appendix A). 

The *B. caapi* samples analyzed in this study were grouped according to place of growth, if the plants were cultivated or native, collecting period (wet season or dry season), presence or absence of swollen stem nodes, and the type and thickness (about 5 or 20 mm) of the liana. In general, THH mean levels were higher in samples collected in the Federal District and Goiás state (4.48 mg/g ± 1.53) when compared with the Amazonian states, but significance was only found with the levels found in plants collected in the state of Acre (1.67 mg/g ± 0.38; Figure 4A). The ratios between THH and the other two ß-carbolines were also significantly different in these groups (data not shown). Harmine mean levels in native plants (5.315 mg/g ± 0.37) were significantly higher than the mean levels found in cultivated plants (3.43 mg/g ± 0.43; Figure 4B), and the mean ratios between harmine and the other two β-carbolines were also significantly higher in native samples (data not shown). No significant differences were found in the β-carboline mean levels among the different types of *B. caapi* (Figure 4C–E). The mean level of total β-carbolines in caupuri samples (5.48 mg/g) was lower than in tucunacá samples (8.03 mg/g), although this difference was not significant. No other significant differences were found for the other investigated parameters and analytes (data not shown).

### 3.2. Quantification of β-carbolines and DMT in Ayahuasca Brew Samples

Table 1 also shows the summary of the alkaloid levels found in the 33 analyzed ayahuasca samples; the results for each sample are shown in Appendix A. Harmaline was found at the lowest concentration (mean of 0.195 mg/mL), and the mean levels of the other alkaloids ranged from 1.08 mg/mL for DMT to 1.28 mg/mL for harmine. The highest variabilities found among the samples were for harmine and harmaline (about 100%). One sample prepared by a Daime group had the highest level of all components (3.12 mg/L DMT, 7.11 mg/L harmine, 0.945 mg/mL harmine, and 3.05 mg/mL THH), and the sample prepared only with *B. caapi* did not contain DMT (Appendix A). The variabilities of the ratios among the β-carbolines ranged from 39.4 to 54.2 (Table 1).

Six ayahuasca samples in this study were prepared using *B. caapi* which sample were also analyzed. The ratios of the β-carbolines were calculated for the ayahuasca samples and for the *B. caapi* extracts, and the results are shown in Table 2. In all samples, the harmaline/harmine ratios were higher in the brew compared to the plant (1.3- to 2.2-fold higher). The THH/harmaline and THH/harmine ratios were higher in the brew in four cases (1 to 4), reaching 10.9- and 6.7-fold higherratios, respectively, in case 3 (Table 2).

## 4. Discussion

In spite of the globalization of ayahuasca use and the substantial number of studies showing a broad spectrum of therapeutic potential of ayahuasca, *B. caapi*, and β-carbolines [13,14,15,16,17,18,19,20,21,22,23,24,25,26,27,28,39], there are very few studies that examined the chemical profile of the beverage and the plant obtained from different sources. The first studies that investigated the levels of β-carbolines in *B. caapi* were conducted in the late 1960s to the 1980s and analyzed 6–7 plant samples collected in the Peruvian Amazon [4,33,34,40]. Schultes et al. [40] analyzed the original material collected by Richard Spruce more than a century before (in 1852) by GC-MS, and the only β-carboline detected in the sample was harmine (at 4 mg/g), which was within the range of the levels found in this study.

The levels of β-carbolines found in the 159 samples of *B. caapi* analyzed in the present study showed a large variability, which was expected as different plant types, native or cultivated, collected during a four-year period in different regions of Brazil were analyzed. The variability among the samples for the total β-carbolines levels was much smaller (40%), indicating that the main differences are in the biotransformation rate among the plants, which affects the individual levels. In the study with the greatest number of *B. caapi* samples analyzed published in the literature (33 samples) [34], the concentration variabilities of β-carboline were smaller compared to the present study (RSDs of 41 to 79%), probably due to the controlled sampling protocol applied in the study (all samples collected between 06:00 to 9:00 a.m. on a single October day in 22 UDV sites). The authors found similar mean concentrations for harmine and harmaline compared to the present study (~4.8 and 0.46 mg/g, respectively), but about half of that for THH (1.0 mg/g).

Samples of native plants analyzed in the present study showed significantly higher harmine content when compared to the cultivated ones, but liana samples of plants cultivated in the Federal District and the state of Goiás (DF/GO) showed a higher THH content when compared with plants from AC, which are all native, except one sample. Various factors are involved in the regulation of plant biosynthetic pathways. We may speculate that the warm and humid forest environment probably accelerates the initiation of β-carboline biosynthesis, and that the cultivated seasonally dry conditions favor the metabolization (reduction) of harmine to THH—a hypothesis that still needs to be further investigated. Furthermore, solar incidence is much lower in native specimens, as the high density of the tropical forest prevents sun rays from reaching the plants beneath and may affect β-carboline biosynthesis. Additionally, DF and GO are both located in the Brazilian savanna biome (cerrado), which is characterized by acidic soil with high aluminum levels and lower organic matter content compared to Amazonian soil [41], which could have an effect on nutrient uptake by the plant and β-carboline biosynthesis. Indeed, the impact of different environmental conditions on the β-carboline biosynthetic rates most likely explain the large variability found in the ratios between the compounds among the *B. caapi* samples, reaching 331% for THH/harmine.

Callaway et al. [34] found slightly, not significantly, higher levels of the β-carbolines in caupuri compared to tucunacá samples, a trend that was not confirmed in the present study. In fact, the mean of total β-carbolines in caupuri samples was lower than in tucunacá samples, although this difference was not significant. Brazilian ayahuasca users describe various subjective and physiological effects when the beverage is prepared with different *B. caapi* types, particularly with caupuri, which is blamed to cause gastrointestinal discomfort. The Peruvian ayahuasqueros also described different physiological effects after drinking beverages prepared with the different types [4], a point that was also mentioned by other authors [34,42]. Most likely, the different effects of ayahuasca observed by the users go beyond the investigated alkaloid profile of the *B. caapi* used and may also be affected by other chemicals in the plants and by the preparation process of the beverage, which will be discussed later in the paper.

Analysis of the 33 ayahuasca samples showed the mean levels of the alkaloids ranging from 0.195 mg/mL for harmaline to 1.28 mg/mL for harmine, with a large variability among the samples for each compound. As expected, one sample prepared only with *B. caapi* did not contain DMT. The traditional use of ayahuasca may not include DMT-containing plants, such as *P. viridis*, as this alkaloid is not the only substance responsible for the effects observed after ayahuasca ingestion [43]. As MAO inhibitors, the β-carbolines affect the levels of neurotransmitters in the central nervous system, including dopamine and serotonin [10,11,12,44], leading to the psychoactive effects sought by the users of *B. caapi*-only beverages.

Most of the studies that investigated the levels of alkaloids in ayahuasca in the last 20 years analyzed fewer than 10 samples, except for Souza et al. (38 samples) [45] and Callaway (29 samples) [43]. Pires et al. [46] analyzed by GC-NPD eight ayahuasca samples prepared by a religious group settled in a Brazilian city and found harmaline concentrations higher than harmine and THH in all samples, which contradicts the results of the present and all other studies conducted with both ayahuasca and *B. caapi* [4,21,22,24,25,43,45,47,48,49].

The highest DMT level found in the ayahuasca samples (3.12 mg/mL in a Daime preparation) was lower than the levels found in two samples provided by the Barquinha and Santo Daime groups and analyzed by Callaway (12.7 and 14.2 mg/mL, respectively) [43], but much higher than the highest level found by Souza et al. [36] in ayahuasca samples provided by UDV (0.34 mg/mL). These differences reflect the different proportions of *P. viridis* and *B. caapi* used in the preparation of the beverage. Indeed, *P. viridis* accounts for 7 to 10% of the plant material in the UDV, while the proportion of *P. viridis* used by the ICEFLU group ranges normally from 17 to 20% (personal communication; Derick Carniello Rezende). Furthermore, the levels of DMT in *P. viridis* also vary substantially among plant specimens and can fluctuate during the day [34].

In the study by Callaway [43], the highest harmine and THH levels were found in ayahuasca samples from the indigenous Shuar people in Ecuador (23–24 mg/mL), but the THH/harmine ratio was similar among all the samples (0.8–1.1). In the present study, this ratio was 1.2 on average but with a much larger variation among the samples (RSD of 331%). Most samples showed a higher concentration of THH when compared to harmine and harmaline, whereas this distribution was the opposite in *B. caapi* samples. This fact could be explained by the possible conversion of harmine to THH (harmine→harmaline→THH; see the structures in Figure 1) through a reduction reaction during ayahuasca preparation, which involves boiling for several hours. Indeed, the harmaline/harmine ratios were higher in all six ayahuasca samples compared to the ratios found in the *B. caapi* used to prepare them, and the THH/harmaline and THH/harmine ratios were also higher in the brew in four samples. The decoction process varies widely among the ayahuasca groups and the ayahuasqueros who prepare it. In general, the plants are boiled in UDV for about 2.5 h, followed by a concentration step of about 3 h before the decoction is ready to be used. This last step varies substantially among other ayahuasca groups, with the liquid being concentrated from 2X up to 9X the initial water volume and can achieve a viscous and sweet material, called Daime honey [50]. These differences in the ayahuasca preparation may indeed impact the β-carboline ratios, although a clear pattern could not be seen in the seven Daime honey samples analyzed in this study (Appendix A). A recent study showed that THH is stable in ayahuasca samples for about 9 days at 37 °C, simulating an extreme transport and storage situation, while harmine and harmaline concentrations vary under the same conditions [51].

The physiological effects of ayahuasca and the pharmacokinetics of its components were investigated using a chemically characterized brew [5,13]. However, the studies did not compare the effects with different chemical profiles. Furthermore, the impact of the different β-carboline concentrations and ratios on the psychological and subjective effects felt by ayahuasca users may be difficult to access. It is quite common that individuals who took the same brew describe different effects, which are recognized as mystical and spiritual experiences [50].

The variability of the chemical profile of the main alkaloids in ayahuasca and *B. caapi* is extremely important and should be considered when interpreting different biological effects found with different materials in animal studies. Da Motta et al. [48] and Oliveira et al. [49] evaluated the developmental effects of ayahuasca on pregnant rats using the same protocol and similar doses, but the results obtained were very different, with the first study showing more important toxicity, including maternal and embryo deaths. The sources of the brew were different in the studies (UDV and Daime, respectively), with different β-carboline and DMT levels and ratios, which probably influenced the observed effects. Furthermore, other components that might be present in the beverages used in the studies could interact and play a role in increasing or decreasing the toxicity. Schwarz et al. [19] tested both *B. caapi* extracts and isolated harmine in MAO inhibition and dopamine release and found the extract to be far more active than the equivalent dose of harmine, suggesting the possibility of synergism among the active compounds present in the plant.

One limitation of this study is that only the main β-carbolines (harmine, harmaline, and THH) were investigated in the *B. caapi* and ayahuasca samples, although various studies report the presence of other harmala alkaloids in the liana and in the brew [11,21,47]. This might have limited the detection of differences in the chemical profile among the various types of *B. caapi* that could explain some of the subjective and physiological characteristics among the types described by ayahuasca users.

## 5. Conclusions

To the best of our knowledge, this is the largest study that analyzed the profile of β-carbolines in native and cultivated *B. caapi* individuals, and the first to report the presence of β-carbolines in *Diplopterys pubipetala*. This large sampling allowed to capture the wide variability of β-carboline profiles in the liana samples, which reflects all the factors involved in the biosynthesis of these compounds under different environmental conditions and organic and inorganic composition of the soil. The analysis of paired ayahuasca/*B. caapi* samples confirmed a previous hypothesis that harmine is reduced to harmaline and further to THH during the preparation of the brew. The data generated in this study should be considered when standardizing plant material for ethnopharmacological uses of *B. caapi* extracts and of ayahuasca. Further studies should be conducted to analyze the other harmala alkaloids in the samples and to correlate the chemical profile of the different types/varieties with the genetic profiles of the plant.

## Figures and Tables

**Figure 1 plants-09-00870-f001:**
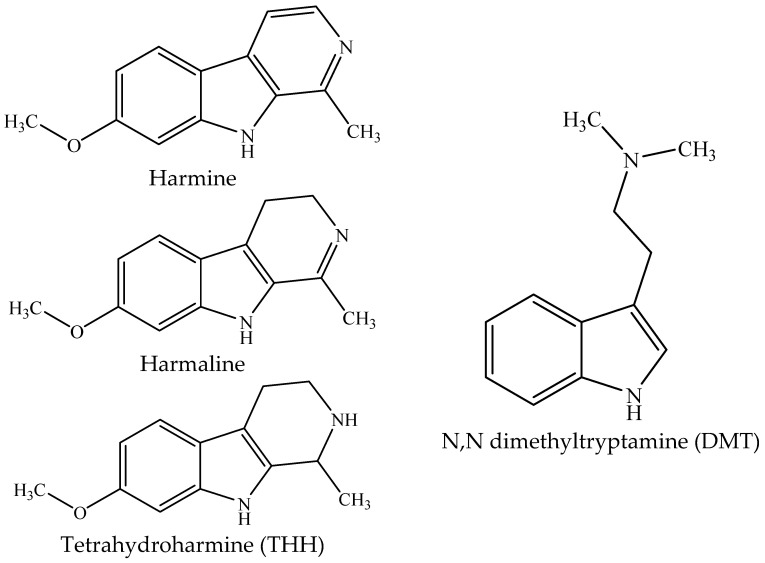
Chemical structures of the β-carbolines harmine, harmaline, and tetrahydroharmine (THH) present in *Banisteriopsis caapi* and ayahuasca and of N,N-dimethyltryptamine (DMT), present in ayahuasca.

**Figure 2 plants-09-00870-f002:**
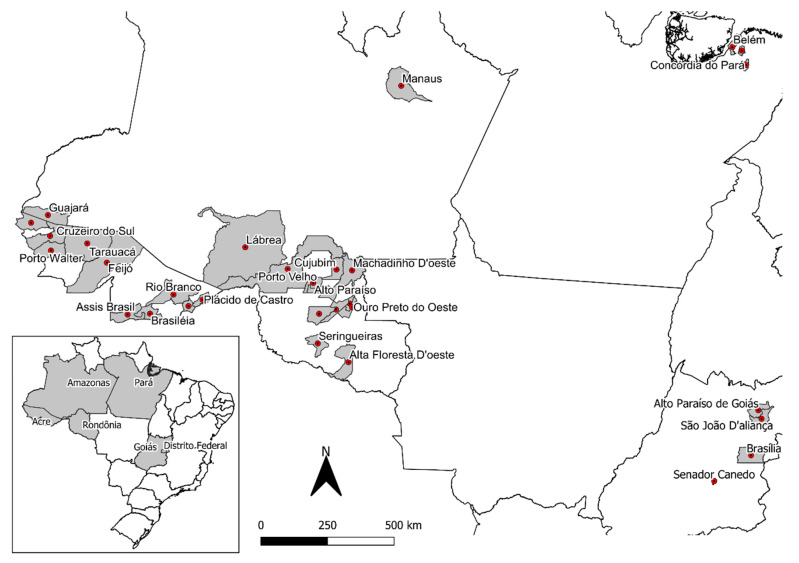
Collection points of *Banisteriopsis caapi* and other species in Brazil.

**Figure 3 plants-09-00870-f003:**
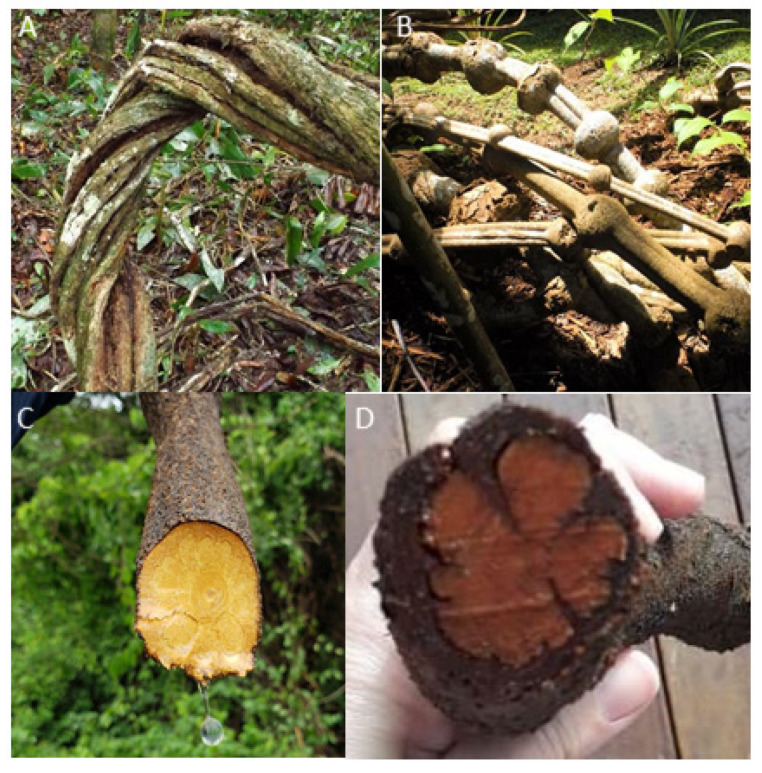
Different *Banisteriopsis caapi* types used in ayahuasca preparation: (**A**) tucunacá; (**B**) caupuri (with swollen nodes); (**C**) fresh amarelinho; (**D**) amarelinho some days after harvesting. Images taken by the authors and collaborators.

**Figure 4 plants-09-00870-f004:**
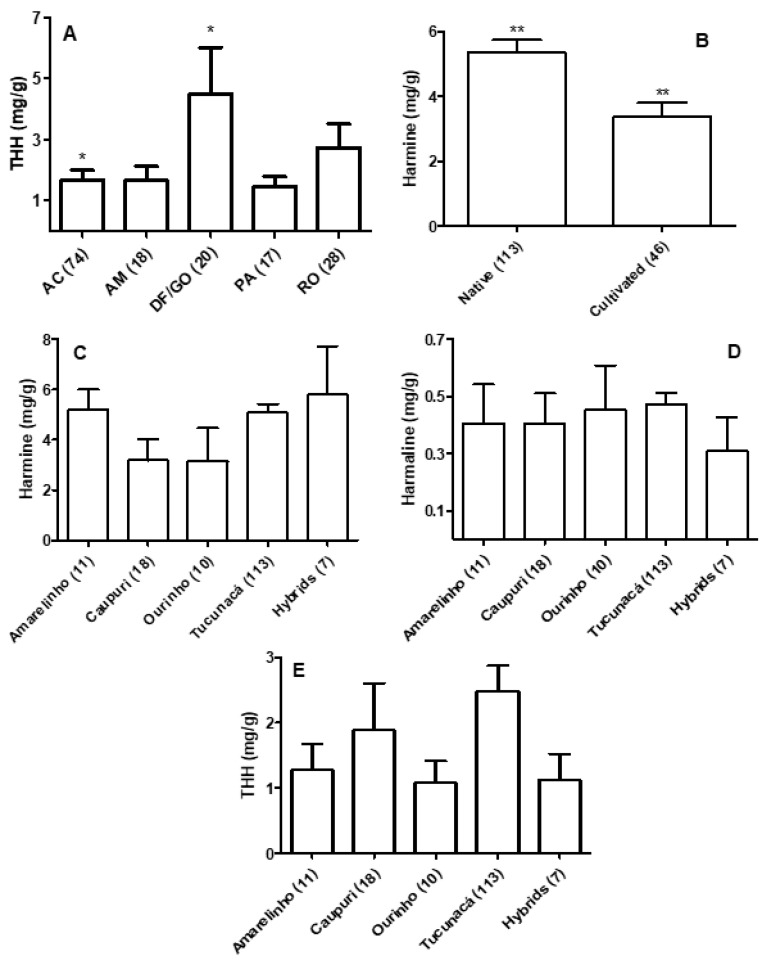
(**A**): THH levels in *Banisteriopsis caapi* samples collected in different regions; two samples collected in the State of Minas Gerais were not included. (**B**): Harmine levels in native and cultivated *B. caapi* samples. (**C**–**E**): β-carboline levels in different *B. caapi* samples. Hybrids include plants described as tucunacá with nodes (2 samples), caupuri without nodes (2 samples), pajezinho (1 sample), caboquinho (1 sample), and quebrador (1 sample). Levels are mean ± standard error of the mean (SEM). * *p* < 0.05; ** *p* < 0.01.

**Table 1 plants-09-00870-t001:** Levels of alkaloids found in *B. caapi* and ayahuasca samples collected in various Brazilian regions from 2016 to 2020.

	*Banisteriopsis caapi* (N = 159)		Ayahuasca (N = 33)	
Compound	Range (Mean)	RSD, %	Range (Mean)	RSD, %
Harmine	<LOD-18.27 mg/g (4.79) ^a^	78.9	0.109–7.11 mg/mL (1.28)	99.1
Harmaline	<LOD-2.08 mg/g (0.451) ^a^	92.4	0.012–0.945 mg/mL (0.195)	102
THH	<LOD-29.04 mg/g (2.18) ^a^	170	0.09–3.05 mg/mL (1.16)	60.3
DMT	-	-	0.10 ^b^–3.12 mg/mL (1.08)	63.8
Total β-carbolines	<LOD-32.1 mg/g (7.4) ^a^	40.4	0.21–11.1 mg/mL (2.65)	76.2
Harmaline/harmine	0.006–3.0 (0.15) ^c^	187	0.04–0.27 (0.155)	39.4
THH/harmaline	0.55–109 (5.6) ^c^	198	2.0–17.0 (8.4)	41.1
THH/harmine	0.01–38 (1.2) ^c^	331	0.35–3.4 (1.2)	54.2

^a^ samples < limit of detection (LOD) were considered as half the LOD (0.0015, 0.0075, and 0.003 mg/g for harmine, harmaline, and THH, respectively); ^b^ does not include one ayahuasca sample that did not contain *P. viridis*; ^c^ only samples with detected levels of all β-carbolines (N = 141). RSD = relative standard deviation.

**Table 2 plants-09-00870-t002:** β-carboline ratios found in six ayahuasca samples and in the respective samples of *B. caapi* used in the preparation.

	Ratio ^a^		
Sample	Harmaline/Harmine	THH/Harmaline	THH/Harmine
Ayahuasca 1	0.255	2.09	8.19
*B. caapi* 1	0.174	0.518	2.99
Ayahuasca 2	0.201	1.63	8.08
*B. caapi* 2	0.153	0.505	3.30
Ayahuasca 3	0.068	0.414	6.12
*B. caapi* 3	0.041	0.038	0.912
Ayahuasca 4	0.188	1.97	10.4
*B. caapi* 4	0.087	0.187	2.14
Ayahuasca 5	0.125	1.64	13.0
*B. caapi* 5	0.094	5.34	56.9
Ayahuasca 6	0.227	1.43	6.27
*B. caapi* 6	0.129	4.52	35.1

^a^ the ratio of the β-carboline in *B. caapi* was estimated from the concentrations in the extract.

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
