# Peer review of "Biodiversity of β-Carboline Profile of Banisteriopsis caapi and Ayahuasca, a Plant and a Brew with Neuropharmacological Potential"

_plants, 2020, doi:10.3390/plants9070870_

Round 1
Reviewer 1 Report
The strengthes of this work are (1.) the interesting topic, because it is an investigation of a traditional beverage („ayahuasca”) containing psychoactive substances, (2.) the very high number of investigated samples not only of beverages but of plants using for establishment, and (3.) the hugeness of the summarized data (results of LC-MS/MS analysis) of the four main components (alcaloids) of these drugs, which can be very useful for other researchers.
The conclusions of this comprehensive work are very temperate. One of the conclusions (highlighted in Abstract as well), that during preparation of ayahuasca harmine is reduced to harmaline and further to THH could be more underline in the text, because there is fully proved by 4 of the total 6 samples (in table 2.) and on the basis of table 1. this conclusion is not confirmed.
I have only one question: According to yours’ lights, have this traditionally used beverage got the potential of pharmaceutical practice? The official pharmacopeia of your country contains it?
I have find only some mistakes. I have signed them in the manuscript.
Author Response
The conclusions of this comprehensive work are very temperate. One of the conclusions (highlighted in Abstract as well), that during preparation of ayahuasca harmine is reduced to harmaline and further to THH could be more underline in the text, because there is fully proved by 4 of the total 6 samples (in table 2.) and on the basis of table 1. this conclusion is not confirmed.
Data on Table 1 refers to all samples and any the discussion we had with the data on Table 2 does not apply. Only six samples (Table 2) had the brew and the plant used to prepare it analyzed, so the discussed about the reduction during ayahuasca preparation can be done only with these samples
I have only one question: According to yours’ lights, have this traditionally used beverage got the potential of pharmaceutical practice? The official pharmacopeia of your country contains it?
As it is largely covered in literature cited in the paper, the plant and the brew have be largely investigated, using both animal and humans subjects. However, they are not yet included in any pharmcaopeia
I have find only some mistakes. I have signed them in the manuscript.
I did not find the manuscript file with your corrections, but the text was revised for the English language and other inconsistencies
Reviewer 2 Report
The manuscript “Biodiversity of β-carboline profile of Banisteriopsis 3 caapi and ayahuasca, a plant and a brew with 4 neuropharmacological potential” by Santos et al. reports the analysis of the chemical profile of B. caapi and of ayahuasca collected in different Brazilian regions. Despite the determination of the chemical composition of B. caapi and ayahuasca has been throughfully studied, it is true that the variability of the chemical profile among different samples from various locations was lacking to the date. Thus, I believe that the work presented in this manuscript is of some interest for the researchers working in this field and I could recomend its publication. However, I want to point out that several important papers are missing in the references section (i.e. doi: 10.3389/fphar.2016.00035, doi:10.1016/j.pharmthera.2004.03.002, 10.3390/molecules25092072). The authors need to revise the literature and make sure that all the important references are included, along with the information therein contained.Author Response
However, I want to point out that several important papers are missing in the references section (i.e. doi: 10.3389/fphar.2016.00035, doi:10.1016/j.pharmthera.2004.03.002, 10.3390/molecules25092072). The authors need to revise the literature and make sure that all the important references are included, along with the information therein contained.
All the suggested references were included in the text. The first two in the introduction section and later, and the third one at the end of the discussion section.
Reviewer 3 Report
Authors investigated the chemical profile of B. caapi and of ayahuas collected in various Brazilian regions. This is a largest study conducted with Malpighiaceae samples and showed a large variability in the main
β-carbolines present in B. caapi and the brews. Overall, the study was well-carried out and will contribute to standardize the material used in ethnopharmacological studies of B. caapi and ayahuasca. However, the following revisions need to be addressed for the manuscript to be considered for publication.
Major Points:
It is difficult to understand Table 2. Firstly, a table should be understandable without referring to Supplementary materials. Therefore, give a short explanation to each samples, for example what the BS136/17 (BS134/17) means. Secondly, it would be much better if authors elaborate what the increases in the ratios of harmaline/harmine, THH/harmine and THH/harmaline in the brew in four case means in terms of the brews' effects to users.
Minors:
- The claim ‘……with neuropharmacological potential’ in the manuscript title is a bit misleading which can be resolved by modifying the title as ‘Biodiversity of β-carboline profile of psychoactive plant Banisteriopsis caapi and brew ayahuasca’.
- Harmine levelin cultivated plants (3.43 mg/kg± 0.43; Figure 4B); the value should be in mg/g.
- Several typographical mistakes and incomplete sentences as highlighted in the manuscript (see attached the file) sometimes make the write up hard to follow and hence should be checked carefully.
- β-carbolines levels in the text need to rechecked with those in the table
- In conclusion, authors stated that ‘…………... reflects all the factors involved in the biosynthesis of these compounds under different environmental conditions’. What factors can be involved should be addressed, provided that seasonal variation is excluded.

Author Response
It is difficult to understand Table 2. Firstly, a table should be understandable without referring to Supplementary materials. Therefore, give a short explanation to each samples, for example what the BS136/17 (BS134/17) means. Secondly, it would be much better if authors elaborate what the increases in the ratios of harmaline/harmine, THH/harmine and THH/harmaline in the brew in four case means in terms of the brews' effects to users.
Table 2 was improved and a paragraph discussing the different ratios and effects was included in the discussion section
Minors:
- The claim ‘……with neuropharmacological potential’ in the manuscript title is a bit misleading which can be resolved by modifying the title as ‘Biodiversity of β-carboline profile of psychoactive plant Banisteriopsis caapi and brew ayahuasca’.
The title was changed according to the suggestion
- Harmine level in cultivated plants (3.43 mg/kg± 0.43; Figure 4B); the value should be in mg/g.
The unit was corrected
- Several typographical mistakes and incomplete sentences as highlighted in the manuscript (see attached the file) sometimes make the write up hard to follow and hence should be checked carefully.
All identified typographical errors and incomplete sentences were corrected
- β-carbolines levels in the text need to rechecked with those in the table.
Indeed, there were some inconsistencies between the numbers in text (Abstract, Section 3.1 and 3.2) and in the Table 1. They were all corrected
- In conclusion, authors stated that ‘…………... reflects all the factors involved in the biosynthesis of these compounds under different environmental conditions’. What factors can be involved should be addressed, provided that seasonal variation is excluded.
An additional information was included in the text